# CODE REPRESENTATION PRE-TRAINING WITH COMPLEMENTS FROM PROGRAM EXECUTIONS

## ABSTRACT

Large language models (LLMs) for natural language processing have been grafted onto programming language modeling for advancing code intelligence. Although it can be represented in the text format, code is syntactically more rigorous in order to be properly compiled or interpreted to perform a desired set of behaviors given any inputs. In this case, existing works benefit from syntactic representations to learn from code less ambiguously in the forms of abstract syntax tree, control-flow graph, *etc*. However, programs with the same purpose can be implemented in various ways showing different syntactic representations while the ones with similar implementations can have distinct behaviors. Though trivially demonstrated during executions, such semantics about functionality are challenging to be learned directly from code, especially in an unsupervised manner. Hence, in this paper, we propose FuzzPretrain to explore the dynamic information of programs revealed by their test cases and embed it into the feature representations of code as complements. The test cases are obtained with the assistance of a customized fuzzer and are only required during pre-training. FuzzPretrain yielded more than 6%/19% mAP improvements on code search over its counterparts trained with only source code or AST, respectively. Our extensive experimental results show the benefits of learning discriminative code representations with program executions.

## 1 INTRODUCTION

Code representation learning is drawing growing attention across the community of artificial intelligence (AI) and software engineering (SE) (Husain et al., 2019; Deng et al., 2023; Liu et al., 2023a). It aims to abstract the structure and the underlying functionality of the source code and embed such semantics into a latent representational space via unsupervised pre-training. By providing general understanding of programs, it is a fundamental task to achieve code intelligence, which enables automated code analysis and processing by fine-tuning with budgeted computation resources or limited human annotations (Rattan et al., 2013; Husain et al., 2019; Lu et al., 2021).

The pre-training recipes (Devlin et al., 2019; Liu et al., 2019) for natural languages have been shown effective in code representation learning (Feng et al., 2020; Radford et al., 2019). However, they neglect that the structure of the code (exhibited by how its elements, *e.g.*, variables and statements, are organized) must comply with precise and rigorous rules. These rules, often referred to as code syntax, are defined by language specification to ensure successful compilation and execution. This inspires recent works to leverage static analysis (Wichmann et al., 1995) to parse and present the structure of code less ambiguously by its syntactic representations, *e.g.* abstract syntax tree (AST) (Guo et al., 2022; Tipirneni et al., 2022) and control-flow graph (CFG) (Allamanis et al., 2018). However, since learning from code as it is a type of static data, existing methods overlook the underlying executable programs. Whilst programs are built with specific purposes of performing a set of behaviors indicated by their functionality, it is challenging to derive such semantics from code structure (Liu et al., 2023a). This is because the same purposes can be implemented by different algorithms in different ways, while the behaviors of programs are susceptible to subtle discrepancies in code. As depicted in Figure 1 (a) and (b), the recursive and iterative insertion sort are notably different regarding their structures, even though they share the same functionality. Meanwhile, the subtle change in Figure 1 (c) that is hard to be noticed can lead to distinct execution results. How to learn discriminative feature representations of code that embed not only the static information from its structure but also the dynamic information about its functionality remains an unsolved problem.

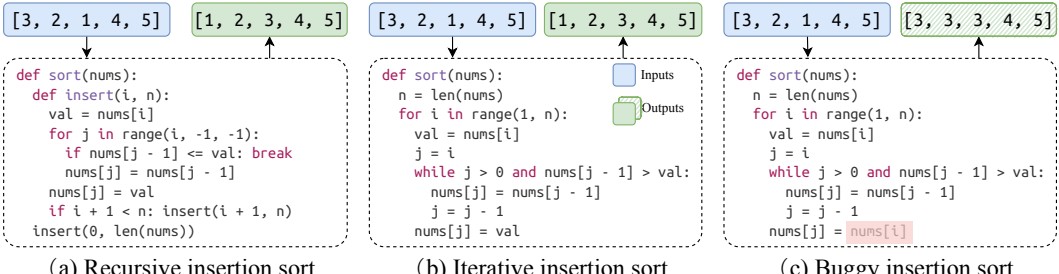

Figure 1: An illustration of implementation variations of the same functional purposes. The source code of **(a)** recursive insertion sort is dramatically different from its **(b)** iterative counterpart regarding their structures, even though their functional equivalence is explicitly demonstrated by the consistent behaviors. On the other hand, the subtle change in **(c)** is buggy and barely observable in comparisons to (b), but can lead to distinct execution results.

In this work, we aim to embed the functional purposes of code into its feature representations, to address the aforementioned limitations of existing works. The key idea is to abstract the behaviors of programs from their input-output relationships (represented by test cases) and enforce the model to infer such information from source code that are readily available in downstream tasks. To achieve this, we take advantage of fuzzing (Zeller et al., 2019) to produce test cases that cover the logic paths of code as comprehensive as possible. Moreover, we propose a novel method called **FuzzPretrain** for joint static and dynamic information modeling. Particularly, in addition to exploring code structure by masked language modeling (Devlin et al., 2021), it formulates a dynamic information matching (DIM) pretext task to tell test cases of different programs apart according to their correspondence to code. By doing so, the model learns holistic feature representations of code and its test cases, encoding both the structure and functionality. FuzzPretrain also involves a self-distillation objective to accomplish a dynamic information distillation (DID) objective. Thereby, the dynamic information is not only properly modelled but distilled from the holistic representations to code features, so to benefit in practice where the test cases are not required.

We make three contributions in this paper: **(1)** We propose to leverage the test cases of programs obtained with the help of fuzzing as explicit indications of functionality to complement their code and syntactic representations. To the best of our knowledge, this is the first attempt to unveil the benefits of fuzzing to code representation pre-training. **(2)** We introduce a novel code pre-training method named FuzzPretrain. It simultaneously models the structure and functionality of code while distilling from such holistic information to represent code in its feature space. It is ready to benefit downstream tasks without extra cost on test cases generations. **(3)** Extensive experiments on four code understanding downstream tasks demonstrate the effectiveness of FuzzPretrain on complementing both source code and its syntactic representations, *e.g.* AST, by dynamic program information for learning discriminative feature representations.

## 2 RELATED WORK

**Code representation learning.** Large language models (Raffel et al., 2020; Liu et al., 2019; Devlin et al., 2021) have achieved unprecedented breakthrough in natural language processing in recent years (Vaswani et al., 2017; Devlin et al., 2021; Radford et al., 2019). Such successes of LLMs have been consistently transferred to code representation learning and advance code intelligence. Early works in the field devote to building large-scale code corpus (Puri et al., 2021; Husain et al., 2019) to be trained simply as plain text at the natural language conventions (Feng et al., 2020; Chen et al., 2021a; Lu et al., 2021). However, the rigorous syntax of programming languages exhibit additional information about code semantics (Hellendoorn et al., 2019; Guo et al., 2021). In this case, recent efforts turn to code-specific designs, *e.g.*, pre-training by identifier detection and infilling (Wang et al., 2021) or parsing the structure of code by its syntax (Guo et al., 2022; Allamanis et al., 2018). Despite being effective, existing approaches consider code as a type of static data and ignore the fact that it is corresponding to an executable program with unique functionality exhibited by its runtime behaviors. Such dynamic information of programs is a critical indicator to tell them apart from others with different purposes but challenging to be inferred from code structures.

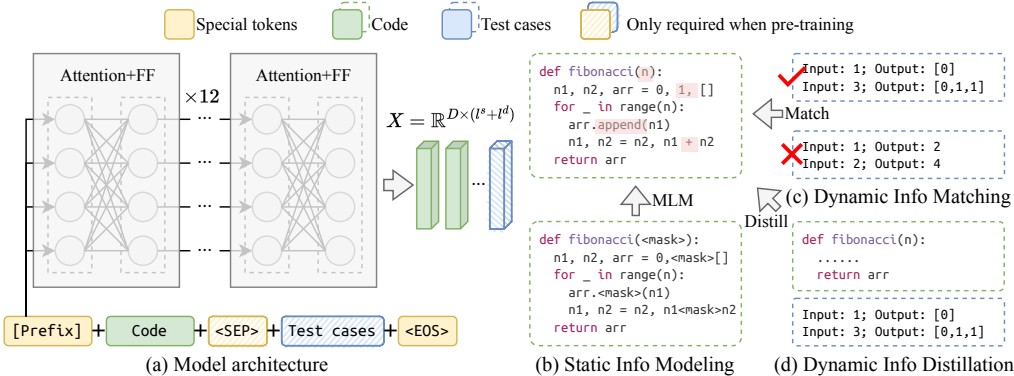

Figure 2: An overview of FuzzPretrain. **(a)** The input sequences are composed of both the code and test cases which are concatenated then encoded by a transformer. FuzzPretrain learns code feature representations by **(b)** static information modeling (SIM) through masked tokens predictions, **(c)** dynamic information matching (DIM) to match test cases to code, and **(d)** dynamic information distillation (DID) to summarize the holistic information about code structure and functionality.

**Static and dynamic code analysis.** Static and dynamic analysis are two common strategies in SE for ensuring the security of software products, and the former has been widely adopted in code representation learning for constructing syntactic representations (Guo et al., 2022; 2021). For early detection of potential vulnerabilities, static analysis (Wichmann et al., 1995) is usually carried out by parsing the structure information of the code for inspections, while dynamic analysis (Khatiwada et al., 2018) detects runtime errors during program executions. Fuzz testing (Zeller et al., 2019), or fuzzing, is a popular dynamic analysis technique, which aims to generate a set of inputs that achieve high code coverage then feed those inputs to programs for execution to observe any unexpected outcome. Overall, the two types of code analysis strategies are usually adopted together as mutual complements to ensure the soundness and completeness of testing.

**Large language models meet software testing.** There are recent efforts on automated bugs mining by LLMs (Schäfer et al., 2023; Kang et al., 2023), which hold an opposite objective to ours on benefiting software testing by code generation. On the other hand, Guo et al. (2021) demonstrate the effectiveness of code understanding from data-flow graph (DFG) extracted via static analysis. Although the intricate dependencies among variables somehow imply the functionality, it suffers from the same problems as the methods introduced by Liu et al. (2023a); Zhao et al. (2023), *i.e.* collecting additional information about structure or functionality of code requires sufficient expertise in SE and this undoubtedly hampers model's applicability when it is required on every downstream task. By contrast, we aim to explore program executions only in pre-training to preserve the benefits of dynamic information to code understanding in practice where test cases are not mandatory. The contemporary TRACED (Ding et al., 2023) model shares a similar insight with ours to benefit code representation pre-training by dynamic program execution. However, we poses an orthogonal solution to them by taking test cases as a type of complementary *data* while they use execution traces for *labels* construction. Beyond TRACED, there are several earlier attempts on training neural networks by dynamic executions. Whilst Wang et al. (2017); Wang & Su (2020); Henkel et al. (2018) explore execution traces for code embedding learning, Shin et al. (2018); Chen et al. (2021b) leverage input-output pairs for program synthesis. In comparison to them, our FuzzPretrain demonstrates the effectiveness of learning code representations from test cases for discriminative code understanding tasks. Test cases are an essential subset of execution traces, which are easier to collect (Shin et al., 2018). More importantly, fuzzing is a language-agnostic process, which vastly reduces the complexity of data collection and is critical for being applied on different programming languages to build code understanding models with multilingual support.

## 3 CODE REPRESENTATION PRE-TRAINING

Given a piece of source code $S$ and a sequence encoder $f_\theta$ parameterized by $\theta$, our objective is to explore the underlying semantics of the code and encode them in a latent representational space

$X^s = f_\theta(S) = \{\boldsymbol{x}_1^s, \boldsymbol{x}_2^s, \cdots, \boldsymbol{x}_{|S|}^s\} \in \mathbb{R}^{k \times |S|}$ in $k$-dimensions. This is to provide general understanding of code, which enables efficient fine-tuning on downstream tasks.

In this paper, we propose to explore the dynamic information obtained during program executions to complement the static information learned from code structure, such that we can embed both in feature representations of code. To that end, we formulate **FuzzPretrain** whose overview is depicted in Figure 2. We first collect a large-scale code corpus based on CodeNet (Puri et al., 2021) and pair each code snippet with multiple test cases synthesized with the assistance of a customized fuzzer. We denote the test cases corresponding to $S$ as $D$ and concatenate it with the code as its joint static and dynamic representation $H = S \oplus D$. By feeding $S$ (or $H$) into $f_\theta$, the features $X^s$ (or $X^h$) are trained by masked tokens predictions (Figure 2 (b)) and test cases to code matching (Figure 2 (c)). Besides, FuzzPretrain distills from the holistic features $X^h$ of code and test cases and embed it into $X^s$, in order to adapt to downstream tasks where test cases $D$ are not available.

## 3.1 FUZZING CODE CORPUS

Fuzzing is a software verification technique that plays an important role in identifying vulnerabilities and enhancing software reliability. A fuzzer verifies the software by repeatedly generating inputs for the software to execute. For each execution, the fuzzer monitors the internal state of the software to determine if the input triggers new behavior. These inputs will be stored for future input generation. Input generation and behavior monitor together allow the fuzzer to effectively focus on exploring new program behaviors. We believe these inputs contain runtime information that cannot be easily found using static analysis. Therefore, using those test cases, *i.e.*, program inputs and corresponding outputs, should supply extra dynamic information to the language model. We employ methods outlined in FuzzTuning (Zhao et al., 2023) to carry out preprocessing, compilation, and fuzzing of the code corpus. In brief, given a C++ or Java code snippet, we use a customized compiler to instrument it before compiling it to an executable. For Python, we modified the interpreter to report the program behaviors during execution. We fuzz the program using AFL++ (Fioraldi et al., 2020) and extract the inputs. Finally, we re-run the program and record the output of the execution. More details about fuzzing existing code corpus can be found in Zhao et al. (2023).

## 3.2 STATIC AND DYNAMIC INFORMATION MODELING

To investigate the versatility of learning with dynamic program information, we build the FuzzPretrain upon two representative models trained on either code or AST, namely CodeBERT (Feng et al., 2020) and UniXcoder (Guo et al., 2022), respectively. We want to emphasize that FuzzPretrain is a generic method that can be integrated into arbitrary static-based models more than the two studied here. FuzzPretrain is ready to benefit different representations of code in a plug-in manner once they are serialized as a sequence of tokens. For clarity, we introduce our designs in terms of source code inputs in this section. The designs for other models like UniXcoder are similar.

**Input/Output representations.** As illustrated in Figure 2 (a), we follow Feng et al. (2020) to concatenate different parts of inputs together with an `<SEP>` token and put an end-of-sentence `<EOS>` token to the end of the concatenation. CodeBERT adopts a begin-of-sentence token `<BOS>` as the prefix for the input sequences while UniXcoder takes `<BOS><ENCODER-ONLY><SEP>` (Guo et al., 2022). For the test cases, we follow Zhao et al. (2023) to decode them from a series of bytes to Unicode strings and then prompt them in a form of natural language to be "Input is: `INPUT`; Output is: `OUTPUT`". This is inspired by how programming online judgement tools (*e.g.* Leetcode[1]) present problem descriptions along with its example test cases to human. We concatenate multiple test cases of a program with the `<SEP>` token to form $D$. We learn from multiple test cases at a time because a single test case is likely to invoke only a part of a program, and only with sufficient number of test cases can we profile the behaviors of the program comprehensively.

We follow the common recipe of natural language processing to split the concatenation of prefix, code, test cases, and suffix as WordPiece (Wu et al., 2016). By feeding the token sequence $S$ into the encoder $f_\theta$, CodeBERT adopts the feature of the `<BOS>` token as its sequence-level representation $\boldsymbol{x}^s$ while UniXcoder applies average pooling on all the tokens to obtain $\boldsymbol{x}^s$.

---

[1]An example of programming online judgement tools: https://leetcode.com

**Static Information modeling.** To learn from the structure of code $S$ according to the dependencies among tokens, we adopt the conventional masked language modeling (MLM) which has been shown simple yet effective on context understanding (Devlin et al., 2021). We follow the common practices to randomly choose $15\%$ of the tokens in $S$ and replace $80\%$ of the selections with a special `<MASK>` token, $10\%$ with random tokens and the remaining are left unchanged. Formally, given the code $S$, a subset $M \subset S$ of it is masked out and leaving a sequence $\tilde{S}$ with replaced tokens. Then, the learning objective is to predict $M$ given the context in $\tilde{S}$:

$$\mathcal{L}_{\text{SIM}}(S) = - \sum_{m \in M} \log p(m|\tilde{X}^s), \tag{1}$$

where $m$ is one of the masked token and $\tilde{X}^s$ is the features of $\tilde{S}$ produced by $f_\theta$. The term $p(m|\tilde{X}^s)$ denotes the probability that $m$ is correctly reconstructed given the incomplete context $\tilde{X}^s$.

**Dynamic Information modeling.** To learn from the dynamic program information obtained during executions, we propose to match the input-output mappings derived from test cases with the functionality inferred from the code. Given a code sequence $S$, we randomly sample an unmatched test cases list $D^-$ and decide whether to concatenate $S$ with its own test cases $D$ or the negative one $D^-$ to form an input sequence $H$ at each training step. We then pair $H$ with a binary label $y \in \{0, 1\}$ indicating whether the mapping relationships embedded in it are consistent. After that, $H$ is encoded by $f_\theta$ to compute its sequence-level representation $\boldsymbol{x}^h$, which is further fed into an additional linear projection layer FC followed by a binary classifier $f_\phi$:

$$\tilde{\boldsymbol{x}}^h = \text{FC}(\boldsymbol{x}^h), \quad \mathcal{L}_{\text{DIM}}(S, D) = \text{BCE}(y, f_\phi(\tilde{\boldsymbol{x}}^h)). \tag{2}$$

In Equation 2, the feature $\boldsymbol{x}^h$ of $H$ is linearly transformed to be $\tilde{\boldsymbol{x}}^h$ and the outputs of the classifier $f_\phi$ indicates how likely the code and test cases in $H$ are matched. Our DIM objective is formulated as a binary cross-entropy loss (BCE) to supervised the predictions from $f_\phi$ by the binary label $y$.

With the supervision of $\mathcal{L}_{\text{DIM}}$, the model is able to derive dynamic information about the execution behaviors of programs and code, given their test cases. However, test cases are not available in many downstream tasks in practice. Therefore, we further devise a dynamic information distillation (DID) objective to simultaneously learn the holistic information from both code and test cases $H = S \oplus D$ and enforce encoding such information in the features of code $S$. Inspired by Tian et al. (2020), we formulate DID in the contrastive learning paradigm to identify the holistic representation $H$ from a list of random samples $H^-$ according to the corresponding source code $S$. To be concrete, we follow He et al. (2020) to maintain a stale copy $f_{\hat{\theta}}$ of the backbone encoder, which shares the identical architecture with $f_\theta$ and is updated accordingly by exponential moving average (EMA) (Lucas & Saccucci, 1990). We then compute the sequence-level feature representation $\boldsymbol{x}^s$ of $S$ and $\hat{\boldsymbol{x}}^h$ of $H$ by $f_\theta$ and $f_{\hat{\theta}}$, respectively. Given the holistic features $X^-$ of a set of random samples $H^-$ computed by $f_{\hat{\theta}}$, which are likely with different semantics from $H$, we train $f_\theta$ to optimize:

$$\mathcal{L}_{\text{DID}}(S, S \oplus D) = - \log \frac{\exp(g(\hat{\boldsymbol{x}}^h, \boldsymbol{x}^s)/\tau)}{\exp(g(\hat{\boldsymbol{x}}^h, \boldsymbol{x}^s)/\tau) + \sum_{\boldsymbol{x}^- \in X^-} \exp(g(\boldsymbol{x}^-, \boldsymbol{x}^s)/\tau)}. \tag{3}$$

The function $g(\cdot, \cdot)$ in Equation 3 computes the cosine similarity between two vectors, and $\tau$ is a temperature hyperparameter controlling the concentration degree of the similarity distribution. In contrast to $\mathcal{L}_{\text{DIM}}$, we always compute the holistic feature $\hat{\boldsymbol{x}}^h$ of code and its own (matching) test cases to avoid the distractions from inconsistent structure and functionality.

**Remarks.** Intuitively, FuzzPretrain can learn dynamic information by masked token prediction of the concatenation of code and test cases $H$. However, our experiment showed that the test cases can be accurately reconstructed in MLM even without the corresponding code in context. We believe that the more valuable information about test cases is their mapping relationships between inputs and outputs, instead of the dependencies between arbitrary tokens. MLM tends to enforce our model to learn all patterns in test cases, which is prone to involving the randomness introduced by fuzzing. This justifies our approach to derive the potential functionality of programs according to the correspondence of code and test cases. More detailed discussions can be found in Section 4.2.

### 3.3 MODEL TRAINING AND INFERENCE

The proposed FuzzPretrain model is optimized alternatively according to static information modeling $\mathcal{L}_{\text{SIM}}$ (Equation 1), dynamic information matching $\mathcal{L}_{\text{DIM}}$ (Equation 2) and dynamic information

Table 1: Evaluations of code representations on code search. Results of our base models (Code-BERT and UniXcoder) are from Guo et al. (2022)'s paper, which are marked in grey because they were not trained on CodeNet. The first and second rows in the header indicate the programming language of the query and the target code snippets, respectively. The column "DYN" indicates whether a model was trained using the test cases or not. mAP scores (%) are reported.

| Model | DYN | Ruby | | | Python | | | Java | | | Overall |
| --- | --- | --- | --- | --- | --- | --- | --- | --- | --- | --- | --- |
| | | Ruby | Python | Java | Ruby | Python | Java | Ruby | Python | Java | |
| CodeBERT | ✗ | 13.55 | 3.18 | 0.71 | 3.12 | 14.39 | 0.96 | 0.55 | 0.42 | 7.62 | 4.94 |
| CodeBERT-MLM | ✗ | 22.45 | 5.67 | 1.95 | 6.74 | 25.70 | 5.01 | 3.61 | 5.84 | 13.45 | 10.05 |
| CodeBERT-MLM+RTD | ✗ | 13.22 | 1.00 | 0.10 | 1.24 | 14.35 | 1.20 | 0.20 | 0.18 | 6.34 | 4.20 |
| **FuzzCodeBERT** | ✓ | **27.92** | **14.88** | **7.92** | **15.39** | **30.47** | **10.26** | **9.94** | **10.65** | **17.75** | **16.13** |
| FuzzCodeBERT w/o DIM | ✓ | 24.05 | 14.08 | 6.96 | 16.32 | 27.51 | 9.54 | 8.66 | 9.76 | 13.49 | 14.49 |
| FuzzCodeBERT w/o DID | ✓ | 18.21 | 2.92 | 0.72 | 2.88 | 25.67 | 3.13 | 0.80 | 1.98 | 17.98 | 8.25 |
| UniXcoder | ✗ | 29.05 | 26.36 | 15.16 | 23.96 | 30.15 | 15.07 | 13.61 | 14.53 | 16.12 | 20.45 |
| UniXcoder-MLM | ✗ | 20.49 | 13.54 | 3.25 | 10.40 | 19.49 | 3.69 | 4.13 | 5.14 | 12.29 | 10.27 |
| UniXcoder-MLM+Contrast | ✗ | 30.83 | 25.73 | 16.46 | 25.44 | 30.50 | 16.80 | 16.01 | 17.26 | 18.86 | 21.99 |
| **FuzzUniXcoder** | ✓ | **42.84** | **29.83** | **17.70** | **33.73** | **47.77** | **21.94** | **20.83** | **23.52** | **33.78** | **30.22** |
| FuzzUniXcoder w/o DIM | ✓ | 22.50 | 13.52 | 6.66 | 15.31 | 22.99 | 6.81 | 7.54 | 6.84 | 12.94 | 12.79 |
| FuzzUniXcoder w/o DID | ✓ | 12.92 | 5.10 | 1.36 | 5.56 | 14.86 | 0.87 | 0.96 | 0.50 | 6.81 | 5.44 |

distillation $\mathcal{L}_{\text{DID}}$ (Equation 3) on each mini-batch of data following (Lample & Conneau, 2019; Guo et al., 2022). At each training step, the stale encoder $f_{\hat{\theta}}$ is updated according to $f_{\theta}$ by EMA: $\hat{\theta} = \lambda\hat{\theta} + (1 - \lambda)\theta$ with a momentum factor $\lambda$, and the holistic representations $\hat{x}^h$ obtained from code and its corresponding test cases will be fed into the queue $X^-$ with the oldest ones inside being removed in a first-in-first-out manner. After pre-training, we keep only the transformer encoder $f_{\theta}$ which is able to yield discriminative feature representations of code $X^s = f_{\theta}(S)$ when only it is available but not the test cases $D$ at inference or on downstream tasks.

## 4 EXPERIMENTS

We collected 1.2M code snippets implemented in C/C++/Python/Java from CodeNet (Puri et al., 2021) to be fuzzed as the training data of FuzzPretrain. We followed the base models, *i.e.*, Code-BERT (Feng et al., 2020) and UniXcoder (Guo et al., 2022) to take a 12-layers transformer with 125M learnable parameters for sequence encoding. We trained FuzzPretrain for 10K steps by the Adam optimizer (Kingma & Ba, 2014), which took around 12/20 hours on 8 Nvidia V100 GPUs for code and AST, respectively. For hyperparameter selections, we carefully aligned with our base models as well as He et al. (2020) regarding $\mathcal{L}_{\text{DID}}$ (Equation 3). We evaluated FuzzPretrain on four standard code understanding benchmarks adopted by Guo et al. (2022) including code-to-code search (abbreviated as code search) on CodeNet, clone detection on POJ-104 (Mou et al., 2016), defect detection on Devign (Zhou et al., 2019) and text-to-code search (abbreviated as text search) on CosQA (Huang et al., 2021). We adopted mean average precision as the evaluation metric for code search and clone detection, accuracy for defect detection and mean reciprocal rank for text search. More details about our implementation and evaluation protocols can be found in Appendix (A.1). Note that test cases are only used in our unsupervised pre-training phase and never used in any downstream tasks in experiments. Code will be made publicly available.

### 4.1 EFFECTIVENESS IN CODE REPRESENTATION LEARNING

**Learning with modality discrepancy.** As our FuzzPretrain model aims to benefit code representation learning with dynamic program information which is often not available on downstream tasks in practice, the first question to be studied here is whether the inconsistency of data modality between pre-training and deployment will refrain FuzzPretrain from benefiting general code understanding. To that end, we adopted the code search task to identify equivalent functions without fine-tuning the code representations or learning additional classifiers. Since our FuzzPretrain models are trained

on different data from the base models (CodeBERT and UniXcoder), we built several fairer baselines to be trained under the exactly same settings as FuzzPretrain but learning from only code or AST without test cases. We presented CodeBERT/UniXcoder-MLM to train by MLM solely as the baselines following Liu et al. (2023b), and CodeBERT-MLM+RTD/UniXcoder-MLM+Contrast to adopt all the losses dedicated to code understanding in their papers for comprehensive exploration on static information modeling. Correspondingly, we denote the two variants of FuzzPretrain built upon CodeBERT and UniXcoder as *FuzzCodeBERT* and *FuzzUniXcoder*, respectively.

As shown in Table 1, the superior performances attained by FuzzCodeBERT and FuzzUniXcoder over their static baselines demonstrate that FuzzPretrain is able to yield discriminative code representations that are beneficial to downstream tasks where test cases are not given. Although Ruby is not one of the programming languages we used for pre-training, FuzzPretrain's improvements on such an unseen language are still on par with that on Python and Java. These results imply that our models did learn to explore code semantics rather than over-fitting to the training data. We attribute the performance superiority obtained by FuzzPretrain to the designs of not only modeling the dynamic information jointly from code and test cases but also distilling such knowledge to be encoded into the feature representations of code. This is evident by the degradation of FuzzPretrain when training without either of the proposed components. Such performance drops further verify the effectiveness of our delicate designs and demonstrate that it is non-trivial to benefit code representation learning by dynamic program information. It is also noteworthy that applying either DIM or DID solely benefits CodeBERT but not UniXcoder. Our hypothesis is that there is a trade-off between the new knowledge acquired and the prior knowledge forgotten during continual learning. It is always easier for the new knowledge to outweigh the prior for a less optimal base model.

**Qualitative studies.** For more intuitive understanding of FuzzPretrain's advantages on code search, we show an example in Figure 3 (a) to exhibit the nearest neighbors of a reference code snippet decided by either UniXcoder or its FuzzPretrain counterpart. It is not surprising that the code with similar structure (*e.g.* variable or function names and the main entry point) can be easily confused with each other by the static-based method even though the false positive example is with different purposes from the reference. On the other hand, such functionality-wise relationships between code snippets are exposed by their execution behaviors, hence, are well captured by FuzzPretrain. To provide a global picture of the learned code features, we adopted t-SNE (Van der Maaten & Hinton, 2008) to visualize the python code for 50 randomly selected problems (classes), which were encoded by the static-based model or our FuzzPretrain in Figure 3 (b) and (c), respectively. The functional equivalence of code

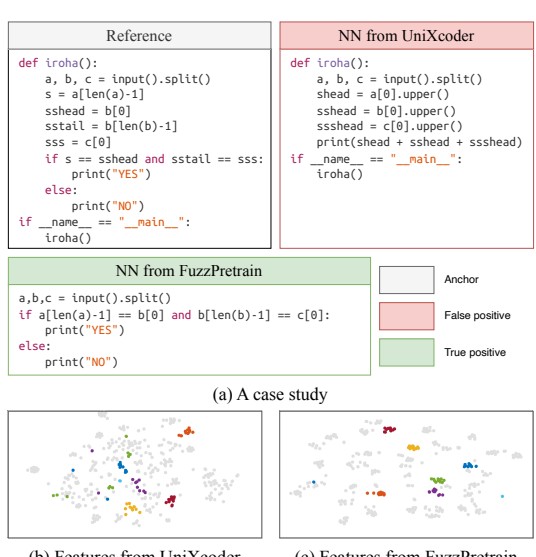

(a) A case study

(b) Features from UniXcoder    (c) Features from FuzzPretrain

Figure 3: Qualitative studies for code search.

are highlighted by the same colors, with only a few random problems are marked with bright colors to avoid chaos. As depicted, the features of functionally equivalent code snippets yielded by UniXcoder can sometimes spread over the feature space sparsely due to their implementation variations. However, our FuzzPretrain forms more compact clusters which are consistent with the underlying semantics of code. These visualizations help explain the potential benefits of jointly learning from the static and dynamic information to comprehensive code understanding.

**Code understanding in novel domains.** We investigated whether our learned code features are transferable and beneficial to downstream tasks in unseen data domains (Lu et al., 2021) in Table 2. We see non-negligible performance advantages obtained by our FuzzPretrain over these static-based methods which learn from only the code structures. The empirical advantage is particularly obvious for CodeBERT which was initially less generalizable, since it learns from code by representing it as a sequence of tokens and overlooks the structure information discovered by static analysis. For UniXcoder, although introducing contrastive learning by feeding the same code inputs to the

Table 2: Comparisons with the static baselines in novel data domains. Results of the base models are marked in grey as training on different data from ours. Results marked with * are reproduced using the checkpoints from authors.

| Model | DYN | Clone | Defect | Text |
|---|---|---|---|---|
| CodeBERT | ✗ | 82.7 | 62.1 | 65.7 |
| CodeBERT-MLM | ✗ | 88.7 | 63.5 | 67.4 |
| CodeBERT-MLM+RTD | ✗ | 84.7 | 62.0 | 66.3 |
| **FuzzCodeBERT** | ✓ | **93.0** | **64.1** | **69.1** |
| UniXcoder | ✗ | 90.5 | 64.5* | 70.1 |
| UniXcoder-MLM | ✗ | 91.2 | 63.8 | 69.8 |
| UniXcoder-MLM+Contrast | ✗ | 91.1 | **65.2** | 69.7 |
| **FuzzUniXcoder** | ✓ | **92.2** | 64.5 | **70.7** |

Table 3: Comparisons with the state-of-the-arts that adopt the same backbone network as ours with 125M parameters. Results marked with * are reproduced using the checkpoints from authors.

| Model (Year) | Clone | Defect | Text |
|---|---|---|---|
| RoBERTa (2019) | 76.7 | 61.0 | 60.3 |
| GraphCodeBERT (2021) | 85.2 | 62.9 | 68.4 |
| DISCO (2022) | 82.8 | 63.8 | - |
| CodeRetriever (2022a) | 88.8 | - | 69.7 |
| ContraBERT (2023b) | 90.5 | 64.2 | 66.7* |
| CodeExecutor (2023a) | 70.5* | 59.0* | 13.1* |
| TRACED (2023) | 91.2 | **65.9** | - |
| **FuzzCodeBERT** | **93.0** | 64.1 | 69.1 |
| **FuzzUniXcoder** | 92.2 | 64.5 | **70.7** |

encoder twice (Gao et al., 2021) (*i.e.*, "Contrast" in Table 2) is helpful to UniXcoder-MLM on defect detection, it leads to subtle performance degradation on the other two tasks. In fact, FuzzPretrain can obtain a similar improvement (from 64.5% to **65.6%**) on the task of defect detection by integrating such a code-to-code contrast (*i.e.*, "Contrast") as well. This further implies the potential of our dynamic information modeling on more advanced base models.

**Comparisons with more state-of-the-arts.** Although FuzzPretrain adopted different pre-training data from the popular bi-modal dataset (Husain et al., 2019) to enable compilation and fuzzing, we compared it with the state-of-the-art models regardless to demonstrate its competitiveness on code understanding. Specifically, we compared FuzzPretrain with three types of methods. RoBERTa (Liu et al., 2019) learns at the natural language conventions. DISCO (Ding et al., 2022), CodeRetriever (Li et al., 2022a), and ContraBERT (Liu et al., 2023b) benefit from contrastive learning as in our solution. GraphCodeBERT (Guo et al., 2021), CodeExecutor (Liu et al., 2023a) and TRACED (Ding et al., 2023) explore program functionality from DFG or execution traces. Note that, GraphCodeBERT extracts DFG from code on every downstream task while CodeExecutor was evaluated without execution traces to better align with practical application scenarios. As shown in Table 3, the performance advantages of FuzzPretrain over GraphCodeBERT implies that mining the functionality of programs from the intricate dependencies among variables is more challenging than modeling from the concrete input-output behavior represented by test cases. Besides, CodeExecutor's inferiority compared to RoBERTa shows that it is difficult to adapt to downstream tasks when execution traces are not available. TRACED is good at code understanding in finer granularity (*e.g.* defect detection) by learning from the detailed internal status of programs in execution traces while our FuzzPretrain is superior on global understanding of code snippets (*e.g.* clone detection) as the test cases we adopted is invariant to implementation variations that are agnostic to functionality. Whilst the methods that are based on contrastive learning of code or its syntactic representations yielded promising results, FuzzPretrain's competitiveness demonstrate the effectiveness of learning with complements from dynamic information. More importantly, FuzzPretrain can be integrated into those methods to further benefit from more advanced modeling of static information.

## 4.2 COMPONENTS ANALYSIS AND ABLATION STUDY

**Effects of different components for dynamic information modeling.** To study the independent contributions of DIM (Equation 2) and DID (Equation 3) to dynamic information modeling, we constructed and compared three variants of FuzzPretrain by removing either or both of them. As shown in Figure 4, the variant of FuzzPretrain trained with only DID (w/o DIM) often out-performed the baselines (MLM) trained with neither DIM nor DID. This indicates that the test cases concatenated after the source code or its syntactic representations potentially play the roles of data augmentation to perturb the distributions of code by supplementing the dynamic information from test cases. Although adopting either DIM or DID is slightly better than FuzzPretrain occasionally, the consistent improvements we brought to different base models on both the retrieval (clone detection) and classification (defect detection) tasks demonstrate the generality of combining the two designs, which is critical for a pre-training method.

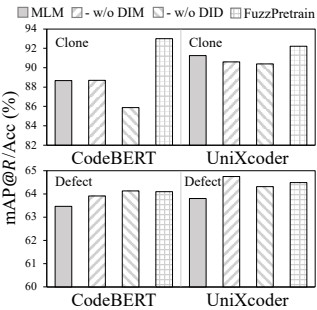
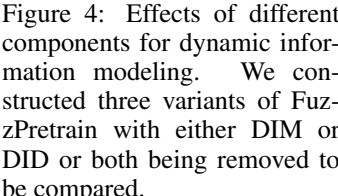

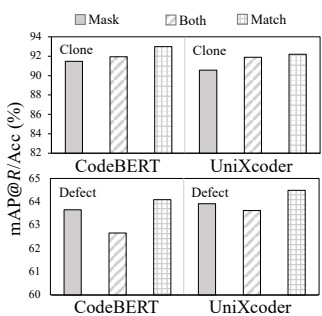

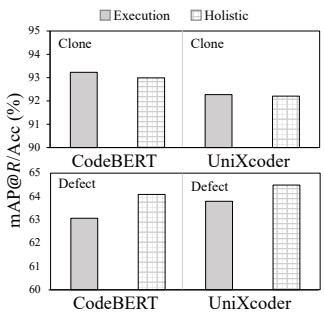

Figure 4: Effects of different components for dynamic information modeling. We constructed three variants of FuzzPretrain with either DIM or DID or both being removed to be compared.

Figure 5: Dynamic information modeling by MLM. The "Mask" variant replaces DIM by MLM for both code and test cases while "Match" is the design we adopted and "Both" is the combination of the two.

Figure 6: Positive pairs in DID. The "Execution" variant constructs the positive pairs in DID using code $T^s$ and its test cases $T^d$, and our "Holistic" design contrasts code to its concatenation with test cases $T^s \oplus T^d$.

**Dynamic information modeling using MLM.** To justify our DIM's effectiveness on dynamic modelling over the conventional MLM, we replace or combine it with MLM on both code and test cases to form two variants of FuzzPretrain as "Mask" and "Both" in Figure 5, respectively. The performance superiority of "Match" to the two variants indicates that applying MLM in test cases is sub-optimal. From our training logs, we observe that the encoder could accurately reconstruct the masked tokens in test cases (or code) regardless of whether the code (or the test cases) is available in the model input. This implies that syntactic and functional representations are both very informative and can be well reconstructed independently, which makes it less straightforward to associate them by MLM. On the contrary, the labels for our DIM is defined only by the relationships between code and test cases, hence, it is infeasible to predict such labels without learning their correlations. Besides, the "Both" alternative tends to associate code with arbitrary patterns in test cases, which are explored by MLM. The resulted correlations can be distracting to code understanding considering the randomness in test cases introduced by fuzzing.

**Positive pairs in DID.** To justify our design of DID, we built a variant of FuzzPretrain which formulates the $\mathcal{L}_{\text{DID}}$ to identify test cases $D$ according to their corresponding code $S$ or AST by constructing the positive pairs in Equation 3 to be $(S, D)$ instead of $(S, S \oplus D)$ in FuzzPretrain. We denote this variant as "Execution" and FuzzPretrain as "Holistic" to be compared in Figure 6. Although the performances of the "Execution" variant on clone detection are on par with that of the "Holistic" counterpart, its inferiority on defect detection is non-negligible. We believe that this is due to the distribution discrepancies between code and test cases (*e.g.* test cases are likely to involve an exhaustive list of random numbers as inputs which are barely seen in code). Therefore, it is more reasonable to jointly learn from test cases and source code to simultaneously benefit from dynamic program information and mitigate the negative impacts from distribution discrepancies.

## 5 CONCLUSION

In this paper, we make the first attempt to derive the program functionality from the dynamic information collected during executions and leverage such knowledge as the complements to the static structure information of code, in order to facilitate comprehensive program profiling and effective code representation learning. To that end, we take advantage of fuzzing to generate diverse and sufficient test cases for a large-scale code corpus efficiently as explicit indications of programs' runtime behaviors. To benefit from such a new modality of data that is often not available on downstream tasks, we proposed FuzzPretrain for joint static and dynamic information modeling. Specifically, FuzzPretrain is trained not only to accomplish the conventional masked tokens predictions objective but also to learn the input-output relationships from test cases encoding the program-specific runtime behaviors, as well as enforcing the model to infer such dynamic knowledge from code structures solely. Extensive experiments were conducted on various code understanding downstream tasks. The notable performance advantages yielded by FuzzPretrain over the models learned from only the structure of code demonstrate the potential benefits of the complements from program executions.

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

# A APPENDIX

## A.1 IMPLEMENTATION, DATASETS AND EVALUATION PROTOCOLS

**Datasets.** Considering the non-negligible time consumption of fuzzing, we collected 1.2M code submissions to $1000/1400/800/250$ problems for C/C++/Python/Java from CodeNet (Puri et al., 2021) to be fuzzed as the training data of FuzzPretrain. We then evaluated FuzzPretrain extensively on four code understanding benchmark datasets: (1) another subset of **CodeNet** (Puri et al., 2021) collected by Guo et al. (2022) consisting of 50K functions implemented in Python, Java, and Ruby for solving one of $4,053$ online coding problems; (2) **POJ-104** (Mou et al., 2016) which contains $104$ C/C++ coding problems with $500$ code submissions to each; (3) **Devign** (Zhou et al., 2019) which is composed of vulnerable functions from four large and popular C-language open-source projects with manual labels; (4) **CosQA** which contains 20,604 pairs of code and real-world web queries (Huang et al., 2021) with annotations from human experts indicating whether the questions raised by the queries can be properly addressed by the code. All the data used for fine-tuning and testing are carefully aligned with previous studies (Feng et al., 2020; Guo et al., 2022).

**Evaluation protocols.** We first investigated the discrimination ability of the learned code representations by code-to-code search (abbreviated as *code search* in the paper) on the subset of CodeNet collected by Guo et al. (2022). In this task, submissions of the same coding problems are assumed to share the same semantics regardless of their implementations. The feature distances between code pairs were adopted to measure their semantic similarity and the mean average precision (**mAP**) was reported to quantify the quality of the retrieval results. We then studied the effects of FuzzPretrain to several downstream tasks in unseen domains, including clone detection, defect detection and text-to-code search (abbreviated as *text search*). The objective of clone detection is similar to that of code search but with fine-tuning in target domains. We followed the same protocol of Feng et al. (2020)'s work to test on POJ-104 and use **mAP@$R$** to assess the results, with only the top-$R$ ($R = 499$) most similar samples were considered in retrieval. In the task of text search, which requires retrieving code snippets according to textual queries, the mean reciprocal rank (**MRR**) is adopted as the metric following Guo et al. (2022)'s work. This evaluation was conducted on CosQA. Defect detection was carried out on Devign and the accuracy (**Acc**) of binary classification is adopted with a fixed threshold of $0.5$.

**Implementation details.** Both our base models, *i.e.* CodeBERT (Feng et al., 2020) and UniXcoder (Guo et al., 2022), followed Liu et al. (2019) to take a 12-layers transformer with 125M learnable parameters for sequence encoding. We followed their designs to set the batch size to 2048 and 1024 while the maximum sequence length to $512$ and $1024$ for CodeBERT and UniXcoder, respectively. In inputs, $400$ and $800$ tokens are reserved for code and AST, respectively, and the rest are for test cases. The test cases of each program were concatenated with the code or the AST by the separation token until reaching the length limits, while the rest was dropped. The FuzzPretrain model was updated by the Adam optimizer (Kingma & Ba, 2014) during training with a learning rate of $2e-5$ for $10K$ steps. For dynamic information distillation $\mathcal{L}_{\text{DID}}$ (Equation 3), we followed He et al. (2020) to set the momentum coefficient $m = 0.999$, the temperature $\tau = 0.07$, and the number of random samples $|H^-| = 2^{16}$. The overall pre-training process took round 12/20 hours on 8 Nvidia V100 GPUs for training with code and AST, respectively. For clarity concern, we summarize the list of notations used in this paper along with their definition in Table 4.

Table 4: A list of notations and their definition.

| Notation | Definition |
|---|---|
| $S$ | A sequence of source code tokens |
| $X^s$ | The token-wise feature representation of $S$ |
| $\boldsymbol{x}^s$ | The sequence-level feature representation of $S$ |
| $D$ | A sequence of test cases tokens |
| $H$ | The concatenated sequence of source code and test cases |
| $X^h$ | The token-wise feature representation of $H$ |
| $\boldsymbol{x}^h$ | The sequence-level feature representation of $H$ |
| $\tilde{S}$ | A sequence of source code tokens with random mask |
| $\tilde{X}^s$ | The token-wise feature representation of $\tilde{S}$ |
| $M$ | A masked subset of tokens |
| $m$ | A masked token |
| $D^-$ | A negative sequence of test cases tokens |
| $\tilde{\boldsymbol{x}}^h$ | The transformed sequence-level feature representation of $H$ |
| $\hat{\boldsymbol{x}}^h$ | The sequence-level feature representation of $H$ produced by $f_{\hat{\theta}}$ |
| $H^-$ | A random set of concatenated code and test cases |
| $X^-$ | The sequence-level feature representations of samples in $H^-$ |
| $\boldsymbol{x}^-$ | The sequence-level feature representation of a negative sample |
| $k$ | The dimension of feature representations |
| $f_{\theta}$ | A deep neural network |
| $f_{\hat{\theta}}$ | A stale copy of $f_{\theta}$ |
| $\oplus$ | The operation for concatenating two sequences |
| $y$ | A binary label indicating whether the source code and test cases concatenated are matching |
| $g$ | The cosine similarity between two feature representations |
| FC | A fully-connected layer |
| $\lambda$ | The momentum hyperparameter in contrastive learning |
| $\tau$ | The temperature hyperparameter in contrastive learning |

Table 5: Evaluations of code representations on inductive code search.

| | c1000 | c1400 | py800 | java250 | Overall |
|---|---|---|---|---|---|
| CodeBERT | 13.95 | 13.22 | 31.23 | 26.72 | 21.28 |
| CodeBERT+MLM | 26.34 | 24.08 | 48.71 | 34.94 | 33.52 |
| FuzzCodeBERT | 69.98 | 68.65 | 78.13 | 69.98 | 71.69 |
| UniXcoder | 17.57 | 15.89 | 55.28 | 45.49 | 33.56 |
| UniXcoder+MLM | 32.84 | 30.28 | 46.79 | 46.90 | 39.20 |
| FuzzUniXcoder | 71.72 | 68.40 | 80.27 | 77.43 | 74.45 |

## A.2 ADDITIONAL EXPERIMENTS AND ANALYSIS

**Inductive zero-shot code search.** We adopted the testing split provided by UniXcoder (Guo et al., 2022) for evaluation of code search, it is likely to overlap with our training data in CodeNet by sharing over 70% of the coding problems. Therefore, we consider the searching of those overlapping samples as transductive inference problems. This is also a practical scenario given that the training data of the latest code models covers a large proportion of open-source projects in Github and is likely to involve the code-of-interests to users. We have also evaluated in an inductive setup where the query and the candidate code snippets are submissions to 50 coding problems of each programming language that have never been seen during pre-training. As shown in Table 5, the superiority of our FuzzPretrain over both the base models and our baselines still holds. That is, these results show that our model is effective not only in the transductive inference setup for code search, but also in an inductive setup where no training/test overlap exist.

## A.3 FUTURE WORKS AND LIMITATIONS

**Fuzzing code corpus.** Our current pre-training data is restricted to OJ-like code corpus (*i.e.*, CodeNet) (Puri et al., 2021), which refrains us from ablating affecting factors in the data distribution in making fair comparison to existing methods. To be more specific, most commonly adopted code corpus (Husain et al., 2019) are composed of standalone functions spread over various software projects (*e.g.*, CodeSearchNet), whose test cases cannot be easily obtained. Whilst OJ data is showing some unique characteristics to benefit our FuzzPretrain model on understanding similar code snippets as indicated by our remarkable performance advantages on POJ-104 (Mou et al., 2016) (Table 3), this also limits our model's generalization ability to other type of code corpus, *e.g.* the F1-score of clone detection on BigCloneBench (Svajlenko et al., 2014) yielded by our FuzzUniXcoder was $1\%$ lower than that by UniXcoder pre-trained on CodeSearchNet. Yet, when both pre-trained on the same selected subset of CodeNet, our FuzzPretrain leads to +0.9% F1 gain in comparison to existing pre-training strategies using, for example, the MLM loss on CodeBERT. Exploring fuzzing on more diverse code corpus help address this limitation.

**Text-code tasks.** Following the discussion about fuzzing code corpus in the previous paragraph, we would like to mention that, since CodeNet does not contain text description of each code, pre-training on it may not fully unleash the power of pre-training on text-code downstream tasks. That is to say, although we have shown the effectiveness of our FuzzPretrain on the text code search task in Tables 2 and 3, even better results can be obtained if we can pair the CodeNet data with text descriptions or if we can pre-train on a dataset with not only texts and code but also test cases. This also withholds FuzzCodeBERT and FuzzUniXcoder from surpassing every state-of-the-art methods on text-code tasks. In addition to exploiting datasets, extensive experiments presented in this paper also verifies complementary effects of dynamic program modeling to these methods, which implies that combining more advanced methods (Wang et al., 2023; Li et al., 2022b) with our FuzzPretrain also leads to superior performance than that of FuzzCodeBERT and FuzzUniXcoder.

**Code generation.** Our designs for dynamic information modeling are all about the holistic comprehensions of code in a global picture, while how to benefit token-wise code understanding by using it is not straightforward. We tested UniXcoder with and without our FuzzPretrain on the python dev split of the line-level code completion task in the CodeXGLUE benchmark (Lu et al., 2021), our FuzzUniXcoder yielded $42.73\%/72.03\%$ Exact Match/Edit Sim *vs.* $42.68\%/71.88\%$ by UniXcoder. We did not observe clear improvements brought by FuzzPretrain on code generation tasks which are usually conducted at token-level, leaving an interesting problem to be studied in the future.

