# OpenReview forum: "Code Representation Pre-training  with Complements from Program Executions"
_ICLR.cc/2024/Conference — ICLR 2024 Conference Withdrawn Submission_

### Official Review · Reviewer_TD6n · 2023-10-24

**Soundness:** 2 fair
**Presentation:** 2 fair
**Contribution:** 2 fair
**Rating:** 3
**Confidence:** 5

**Summary:**

The paper focuses on improving code intelligence in Large Language Models (LLMs). Recognizing the challenges of learning from code's semantics and varied implementations, the authors introduce "FuzzPretrain." This approach uses dynamic program test case data, obtained via a custom fuzzer, during pre-training to enrich code feature representations. Compared to models trained with just source code or AST, FuzzPretrain achieved improvements in code search performance, emphasizing the value of integrating program execution insights.

**Strengths:**

incorporating program executions to pre-train LLMs is new despite a narrow and low-level contribution.

**Weaknesses:**

- My biggest concern is the key idea of using program execution to learn code models is just not new. Authors seem to be completely unaware of the vast literature of neural code models based on dynamic executions. Here are some papers for authors' reference:

  1. dynamic neural program embedding for program repair
  2. blended precise semantic program embeddings
  3. Improving Neural Program Synthesis with Inferred Execution Traces
  4. Latent Execution for Neural Program Synthesis Beyond Domain-Specific Languages
  5. Code vectors: understanding programs through embedded abstracted symbolic traces

   Even though they do not directly target LLM rightfully so given that LLMs are not even around at the time those papers are published, their works already share the insight of how dynamic execution can benefit learning of code embeddings. Therefore, it's entirely inappropriate for authors to totally ignore them.

- The evaluation task of clone detection is poorly handled. First and foremost, clone detection is almost an entirely syntactic task as tools are asked to detect the syntactic similarity of code, however, incorporating execution traces are semantic information that totally ignores the syntactic features. So this clone detection task does not even match the insight of the paper. Of course, I am aware of the type 4 semantic clones, however, the question is, how many are those in POJ-104, authors provide no information in this regard, and it's more than reasonable to assume there are very few if any used in the evaluation.

- In ablation study, Fig 3 demonstrates that in Defect DIM and DID is not necessary because removing them actually yields a bigger gain over FuzzPretrain. This casts doubt on the effectiveness of your technique.

**Questions:**

See weaknesses

---

> ### Author Response · Authors · 2023-11-20
> **Responses to Reviewer TD6n**
>
> W1: authors seem to be completely unaware of the vast literature of neural code models based on dynamic executions
> **ANS:** Thanks for suggesting these works on training neural networks by program executions. We have added discussions about these works in our paper. Indeed, [1,2,5] explore execution traces for code embedding learning and [3,4] leverage input-output pairs for program synthesis. In comparison to them, we demonstrate the effectiveness of learning discriminative code embeddings from input-output pairs (test cases) for code analysis/understanding. Test cases are an essential subset of execution traces, which are easier to obtain [3]. More importantly, by exploiting test cases, we can take advantage of fuzzing which is almost a language-agnostic process to vastly reduce the complexity of data collection and make it easy to be applied to different programming languages to build code understanding models with multilingual support.
>
> W2: The evaluation task of clone detection is poorly handled. incorporating execution traces are semantic information that totally ignores the syntactic feature
> **ANS:** We appreciate the comment, but there seems to be a misunderstanding. We didn't involve any execution information when fine-tuning on the clone detection task. In fact, one of the key contributions of our work is to ensure that only the source code (or its syntactic representations) is needed for downstream code understanding tasks (including clone detection) while still yielding competitive performances when compared to Zhao et al's work [6]. Specifically, Zhao et al explicitly use test cases for tuning/testing in the clone detection downstream task and achieve 92.01% mAP on POJ-104 while our method achieves 93.0% without incorporating execution traces as semantic information when performing clone detection, both adopting the CodeBERT as the base model. We have reiterated this at the beginning of our experiment section to make this more explicit.
>
> W3: Fig 3 demonstrates that in Defect DIM and DID is not necessary because removing them actually yields a bigger gain over FuzzPretrain
> **ANS:** Applying either DID or DIM solely may outperform our FuzzPretrain occasionally to a very limited extent. We hypothesise this is due to our supervision signal from joining DIM and DID for global code modelling in sequence-level is sometimes overstrong. This is likely to dominate our continual learning of the base models which were pre-trained with token-wise objectives to be sensitive to local fine-grained features revealing defects. However, we would like to argue that the general performance advantages yielded by FuzzPretrain when being applied to different base models on different tasks demonstrate not only its effectiveness but also its generality, which is critical for a pre-training method. It is also the reason why pre-trained models should be tested on several different tasks to demonstrate their effectiveness.
>
> References
> [1] Ke Wang, Rishabh Singh, and Zhendong Su. Dynamic neural program embedding for program repair. arXiv preprint arXiv:1711.07163, 2017
> [2] Ke Wang and Zhendong Su. Blended, precise semantic program embeddings. In Proceedings of the 41st ACM SIGPLAN Conference on Programming Language Design and Implementation, pp. 121–134, 2020
> [3] Eui Chul Shin, Illia Polosukhin, and Dawn Song. Improving neural program synthesis with inferred execution traces. Advances in Neural Information Processing Systems, 31, 2018
> [4] Xinyun Chen, Dawn Song, and Yuandong Tian. Latent execution for neural program synthesis beyond domain-specific languages. Advances in Neural Information Processing Systems, 34: 22196–22208, 2021
> [5] Jordan Henkel, Shuvendu K Lahiri, Ben Liblit, and Thomas Reps. Code vectors: Understanding programs through embedded abstracted symbolic traces. In Proceedings of the 2018 26th ACM Joint Meeting on European Software Engineering Conference and Symposium on the Foundations of Software Engineering, pp. 163–174, 2018
> [6] Jianyu Zhao, Yuyang Rong, Yiwen Guo, Yifeng He, and Hao Chen. Understanding programs by exploiting (fuzzing) test cases. ACL, 2023

---

> > ### Author Response · Authors · 2023-11-22
> > **Any further questions or comments?**
> >
> > Dear Reviewer TD6n,
> >
> > We are truly grateful for your review and hope that our responses have effectively addressed your concerns. Specifically,
> > + we have discussed the suggested related works and spelt out our contribution over them on exploring test cases for discriminative code representation learning;
> > + we have clarified a potential misunderstanding that we didn't involve any execution information on clone detection, and showed our competitiveness to the model that explicitly adopted executions for this task;
> > + we have explained the observation of our experiments on defect detection and the necessity of both DIM and DID which ensures the generality of our FuzzPretrain to different downstream tasks.
> >
> > As the final deadline for reviewer-author discussions draws near, we are eager to receive any additional feedback you may have. We are more than willing to offer further clarification and engage in a deeper conversation if any of your concerns remain. Many thanks.
> >
> > Best regards,
> > Authors

---

### Official Review · Reviewer_2cyZ · 2023-10-28

**Soundness:** 3 good
**Presentation:** 4 excellent
**Contribution:** 2 fair
**Rating:** 6
**Confidence:** 4

**Summary:**

This paper proposes FuzzPretrain which leverages fuzzing to derive input/output test cases for doing continued pre-training on CodeBERT and UniXcoder. Their pre-training strategy encompasses three objectives: (1) Static information modeling: learning from the sequence representation of code using MLM, (2) Dynamic information modeling: learning to distinguish between the positive test cases and randomly sampled negative test cases for a given code snippet, using binary cross-entropy loss, and (3) Dynamic information distillation: learning to position the code representation corresponding to the code snippet alone near to the representation corresponding to the code snippet concatenated to the test cases, using contrastive learning with randomly sampled negative code representations.They conduct this pretraining using 1.2M codes snippets in C/C++/Python/Java from CodeNet.  They evaluate their approach (without any further fine-tuning) on four downstream code understanding tasks: code search, clone detection, defect detection, and text-to-code search. Results demonstrate improvements over baselines which use only static information.

**Strengths:**

- This paper proposes leveraging fuzzing for pre-training which may inspire future techniques for building better pre-training datasets and objectives for code models.
- The approach proposed by this paper yields impressive improvements for code search and smaller improvements across three other code understanding tasks, relative to comparable models which use only static information.
- The pre-training tasks that the authors propose are quite interesting, and the extensive analyses and ablation studies that are included in the paper are helpful for understanding the contributions of these tasks.
- The paper is very well-written.

**Weaknesses:**

- The authors seem to suggest novelty in using dynamic program information for learning code representations through claims like "To the best of our knowledge, this is the first attempt to unveil the benefits of dynamic program information on code representation pre-training" and "...we make the first attempt to derive the program functionality from the dynamic information collected during executions." However, there is work that does similar things, one of which they have cited, and others that they have cited. Namely, they have not cited "TRACED: Execution-aware Pre-training for Source Code" (https://arxiv.org/pdf/2306.07487.pdf) leverages dynamic information, specifically executable inputs and corresponding execution traces, for pre-training. Though fuzzing is not used there, it is used for building code representations in a paper that is cited: "Understanding Programs by Exploiting (Fuzzing) Test Cases" (https://arxiv.org/pdf/2305.13592.pdf), though they are not actually using them for pretraining. It seems that the contribution is more around using specifically fuzzing for pre-training. I believe this should more clearly be conveyed.
- Related to the previous point, they present results for an approach that does use dynamic information for pre-training: CodeExecutor (https://arxiv.org/pdf/2305.05383.pdf). However, they do not present this on the main code search task in Table 1 (which is also more aligned with what CodeExecutor was actually benchmarked on). In fact, many of the "state-of-the-art" models listed in Table 3 are not included in Table 1 for code search. It is not clear why these results were excluded from the paper. The same goes for the analyses in Figures 4-5. Since much of the focus was on code search, the ablations and analyses would be based on that.
- As the authors themselves acknowledge in the limitations section, this work is focused on code understanding tasks and no generative tasks. I find this a bit troublesome because the underlying model that is used, UniXcoder, was originally designed to also handle generative tasks. CodeExecutor was also benchmarked on code generation. The authors do not report results for generative tasks like code generation or summarization.

**Questions:**

1) Please address the points made above.
2) Please provide additional details on fuzzing. Namely, on average, how many test cases are generated using fuzzing for each code snippet? On average, how long does it take to generate test cases for each example? How scalable is this technique for a much larger pre-training corpus?
3) Some parts of the DIM objective are a bit unclear to me. Namely, it seems that both the inputs and outputs are included in the test cases. It seems that a model could learn to exploit just the structure of the input rather than reasoning about the output because of this, especially since negative examples are sampled randomly from the whole corpus. More concretely, suppose you have a code snippet that takes in a list of integers and the expected functionality is returning the product of these integers. So, a positive test case would be (input: [5,3,2], output: 30) and a negative test case corresponding to another snippet could be (input: "hello", output: "Hello"). If the objective is trained using such examples, then the model may learn to just exploit surface patterns like the input being a list of integers versus the input being a string. To ensure that the model is actually reasoning about runtime behavior, it seems that you have to force the model to reason about the output (e.g., masking the output) or do hard negative example mining. Could you clarify this?


Other suggestions:
- Please fix the capitalization in paper titles in the appendix (e.g., Graphcodebert $\rightarrow$ GraphCodeBERT, Coderetriever $\rightarrow$ CodeRetriever)
- Beginning of Section 3.2, "presentations of code" $\rightarrow$ "representations of code"

---

> ### Author Response · Authors · 2023-11-20
> **Responses to Reviewer 2cyZ**
>
> W1: there is work that does similar things
>
> + they have not cited "TRACED..." leverages dynamic information, specifically executable inputs and corresponding execution traces, for pre-training
> **ANS:** Thanks for suggesting this contemporary work. We have discussed this work in our updated paper. Here, we'd like to clarify our difference is that we explored test cases as another modality of *data* in inputs while they use execution traces to build *labels* for learning their model's outputs. We believe these are two orthogonal attempts of a similar idea and are not contradictory. Specifically, our FuzzPretrain showed superior performance on clone detection in POJ-104 to TRACED by synthesising comprehensive test cases to provide a holistic picture of the functionality of code snippets, while they are better on defect detection by explicitly learning from the finer-grained internal status in execution. Since they are orthogonal, the two methods can possibly to combined to achieve even better results.
> Some initial comparisons and discussions have been added to the related work section and Table 3, respectively.
>
> + "Understanding Programs by Exploiting...", though they are not actually using them for pretraining
> **ANS:**
> We would like to stress that, as discussed in Section 2 in our paper, we dedicatedly designed our model to avoid requiring additional information (i.e., test cases) about code in downstream tasks, and this tells our difference to the suggested work. Test cases are only used in the unsupervised pre-training phase and never used in downstream tasks in experiments in this paper. We have revised the paper to make this point clearer.
>
> W2: they present results for an approach that does use dynamic information for pretraining: CodeExecutor...they do not present this on the main code search task in Table 1. many of the "state-of-the-art" models listed in Table 3 are not included in Table 1 for code search. The same goes for the analyses in Figures 4-5
> **ANS:** This is because our results in Table 1 and Figures 4-5 are NOT for comparisons to the state-of-the-art but to provide in-depth analyses of our model for its comprehension. Therefore, we included only the models that trained on the same data as our FuzzPretrain to conduct fair comparisons and derive reliable conclusions, in addition to demonstrating our competitiveness to the state-of-the-art that trained on different data in a separate table (Table 3). We further evaluated CodeExecutor on the code search task in Table 1 without re-ranking by execution traces. It yielded 3.54% overall mAP vs 16.1% and 30.2% by our FuzzCodeBERT and FuzzUniXcoder, respectively. These results imply that the CodeExecutor seems to overfit to the execution tasks while not being able to produce discriminative feature representation for source code directly. This is consistent with the results we reported in Table 3.
>
> W3: the underlying model that is used, UniXcoder, was originally designed to also handle generative tasks. The authors do not report results for generative tasks
> **ANS:** We evaluated UniXcoder with and without FuzzPretrain on the python dev split of the line-level code completion task from CodeXGLUE. For fair comparisons, we also report the results of our MLM baseline which was trained on the same CodeNet data as ours without using the generated test cases. As shown in the below table (results in which have also been added to the appendix of the paper), the benefits brought by FuzzPretrain are small but consistent. This is because our designs for dynamic information modelling are all about learning holistic representation of complete code snippets to be discriminative, regarding their functionality but not token-level generation. This contributes the most to code understanding and does not directly benefit code generation. However, we believe that, as a seminal work of incorporating test cases into programming language model pre-training, our work will inspire future methods which could improve code generation by taking advantage of test cases in pre-training.
>
> | Model         | DYN | Exact Match    | Edit Sim |
> |---------------|-----|----------------|----------|
> | UniXcoder     | No  | 42.68          | 71.88    |
> | UniXcoder+MLM | No  | 42.68          | 71.67    |
> | FuzzUniXcoder | Yes | 42.73          | 72.03    |

---

> > ### Author Response · Authors · 2023-11-20
> > **Responses to Reviewer 2cyZ**
> >
> > Q1: Please address the points made above
> > **ANS:** Please see W1 - W3.
> >
> > Q2: Please provide additional details on fuzzing. how many test cases are generated using fuzzing for each code snippet? how long does it take to generate test cases? how scalable is this technique for a much larger pre-training corpus?
> > **ANS:** We generated around 20 test cases for each code snippet and it took 20k CPU hours to fuzz the whole code corpus composed of 1.2M code samples (approximately 10 days with 80 CPUs in parallel). The fuzzing process is completely automatic and is ready to scale up to a larger corpus via parallelisation and implementation on, for example, GPUs/TPUs.
> >
> > Q3: To ensure that the model is actually reasoning about runtime behavior, it seems that you have to force the model to reason about the output (e.g., masking the output) or do hard negative example mining.
> > **ANS:** Thank you for the actionable suggestion. We further added a masked language modelling objective for the outputs in test cases to our FuzzUniXcoder for their prediction. From the results reported in the below table, we didn't observe clear and consistent improvements brought by such a design. We suspect that this is because the inputs to different programs also have complex patterns which makes them less trivial to match with source code. To verify this, we count the variable types in the inputs to each python program and found that more than 87% of the programs mix numeric and non-numeric variables in their inputs.
> >
> > |                                     | Clone Detection | Defect Detection | Text Search |
> > |-------------------------------------|-----------------|------------------|-------------|
> > | FuzzUniXcoder                       | 92.2            | 64.5             | 70.7        |
> > | FuzzUniXcoder w/ Outputs prediction | 91.7            | 64.7             | 69.9        |

---

> > ### Author Response · Authors · 2023-11-21
> > **Responses to Reviewer 2cyZ**
> >
> > Regarding W2, in addition to CodeExecutor, we further tested ContraBERT on the code search task in CodeNet, which yielded 8.15% overall mAP with the CodeBERT as their base model while our FuzzCodeBERT's is 16.13%. Although we both adopted contrastive learning for code representation learning, our superiority to ContraBERT further evidences the effectiveness of our DID and the benefits brought by dynamic program information.

---

> > > ### Comment · Reviewer_2cyZ · 2023-11-21
> > >
> > > Thank you for providing detailed answers to my questions and for providing additional results. I have read the other reviews and responses to the other reviews. I will be keeping my score the same.

---

### Official Review · Reviewer_EByC · 2023-11-01

**Soundness:** 3 good
**Presentation:** 2 fair
**Contribution:** 2 fair
**Rating:** 5
**Confidence:** 4

**Summary:**

This paper proposed FuzzPretrain to incorporate program dynamic information to code pre-trained models to supplement the missed information. Specifically, FuzzPretrain utilized a customized fuzzer to obtain some testcases of a code snippet and then utilize the testcases as well as the original code snippets for pre-train. It designed three pre-training objectives, in addition to MLM, dynamic information matching (DIM) is designed to separate test cases of different programs and dynamtic information distillation objective is used to learn the holistic information from the code and test cases. Four downstream tasks including code-to-code search, clone detection, defect detection and text-to-code search are used to evaluate the model effectiveness. Furthermore, an ablation study is also conducted to illustate the effectiveness of their proposed pre-training objectives.

**Strengths:**

+The idea to incorporate program dynamic behaviors in pre-training seems ok to supplement current code pre-trained models.
+This paper is easy to follow and understand.

**Weaknesses:**

-The technique novelty is lack. I agree that the program dynamic information is important and can benefit code pre-trained models, however in this paper, the usage of dynamic information is too easy. It just concatenates the test cases with the original code for model training. I am not sure how much dynamic information is contained in the test cases. Furthermore, I am confused that why test cases are enough for using dynamic information? Lastly, is there any other way to use dynamic information rather than such a simple way?

-In terms of model design, the novelty is limited. It uses BERT-style model as the model architecture for pre-training, why not use more powerful encoder-decoder model and there are some works such as CodeT5[1] and CodeT5++[2] have proved encoder-decoder is better than CodeBERT. Furthermore, the designed pre-trained tasks DIM and DID are also simple, DID is similar to InfoNCE[3].

-In terms of downstream tasks, the evaluation tasks are also limited. There are only four downstream understanding code tasks, more code-related tasks are need to evaluate to confirm the effectiveness of the proposed approach.

[1] Wang et al. CodeT5: Identifier-aware Unified Pre-trained Encoder-Decoder Models for Code Understanding and Generation.

[2] Want et al. CodeT5+: Open Code Large Language Models

[3] Liu et al. Contrabert: enhancing code pre-trained models via contrastive learning

**Questions:**

-Why use test cases to represent code dynamic information, can test cases are sufficient to represent code dynamic information?

-How to construct negative samples in DID pre-training task at equation 3? Please give more explanation?

-Can FuzzPretrain wok well in some code generation tasks such as code summarisation, code completion?

---

> ### Author Response · Authors · 2023-11-20
> **Responses to Reviewer EByC**
>
> Q1: Why use test cases to represent code dynamic information, can test cases are sufficient to represent code dynamic information?
> **ANS:** Test cases provide explicit indications of programs' functionality without any distractions from implementation variations, e.g. the data-flow graph or execution trace of the same functionality might vary depending on its implementation. They reveal programs' execution behaviours and are sufficiently informative to support challenging code-related tasks like code generation [1,2]. More importantly, test cases are easier to collect and thus more scalable for different programming languages to enable multilingual code understanding.
> We added the comparisons to a contemporary method [3] which was trained using execution traces. It turns out that our FuzzPretrain is superior on the clone detection task which evaluates the code feature's sensitivity to syntax variations of the same functionalities, i.e. our FuzzCodeBERT yielded 93.0% mAP vs 91.2 by TRACED [3]. Detailed results and discussions about this can be found in the revised Table 3 in our paper.
>
> Q2: How to construct negative samples in DID pre-training task at equation 3? Please give more explanation.
> **ANS:** The negative sample set is constructed randomly. As illustrated in Section 3.3, we simply followed He et al [4] to feed all the samples in a mini-batch into a queue and remove the oldest one inside when the queue is full in each forward pass. We then take the queue as the negative sample set in the next iteration. Given that our mini-batch is constructed randomly, the negative sample set is also random in every iteration. More sophisticated methods may lead to even better results.
>
> Q3: Can FuzzPretrain wok well in some code generation tasks such as code summarisation, code completion?
> **ANS:** As stated in Appendix A.3 of our updated manuscript, we tested but didn't observe notable improvements brought by FuzzPretrain to the base models on generative tasks. To be concrete, we yielded 42.73%/72.03% Exact Match/Edit Sim on the python dev split of the line-level code completion task in CodeXGLUE vs 42.68/71.88 by UniXcoder. We hypothesise this is due to all of our designs for dynamic information modelling encouraging holistic comprehensions and discrimination of code snippets but not on token-wise generation.

---

> > ### Author Response · Authors · 2023-11-20
> > **Responses to Reviewer EByC**
> >
> > W1: The technical novelty is lack
> >
> > + it just concatenates the test cases with the original code for model training
> > **ANS:** We argue that it is non-trivial to benefit from test cases and we did much more than a simple concatenation for this sake. We conducted throughout experiments in Table 1&2 as well as the ablation studies to explicitly demonstrate that trivially applying common training objectives (e.g., MLM or InfoNCE in contrastive learning) on the concatenation of code and test cases is suboptimal to benefit code understanding. For example, we demonstrated in Figure 5 that source code and test cases are too independently informative to be associated by MLM. We outperformed our ``Mask'' counterparts, which used MLM instead of our DIM for dynamic information modelling, by **1.5%/1.6%** on clone detection when adopting CodeBERT and UniXcoder as the base models, respectively.
> > Similarly, by comparing to the different variants of contrastive learning in Tables 1&2 as well as Figure 6, we show the complements from functional semantics to syntax features, which is more effective than learning from either independently.
> >
> > + is there any other way to use dynamic information rather than such a simple way
> > **ANS:** Whilst this question is open-ended, we did explore several different ways to learn from test cases by the different variants we built in Figures 5&6. For example, the "Mask" variant in Figure 5 applied MLM on the concatenation of code and test cases while the "Execution" variant in Figure 6 conducted code-to-test cases contrastive learning. The comparisons to these variants justify our model designs on effective dynamic program information modelling, and more importantly, benefitting source code understanding.
> > More sophisticated ways of utilising dynamic information also include utilising execution traces [3,5], while, as discussed in our response to Q1, our method can be both effective and more scalable.
> >
> > W2: In terms of model design, the novelty is limited
> >
> > + why not use more powerful encoder-decoder model
> > **ANS:** The UniXcoder we adopted as one of our base models can actually act as an encoder-decoder model by prefixing inputs with the `<encoder-decoder>` token and using casual attention masks [6]. As we highlighted, we focused on code understanding tasks and all our training objectives are discrimination-based, so mostly applied to encoder models. Due to the time limit in the rebuttal period, experiments on more recent encoder-decoder models (e.g., CodeT5+) are considered as future work.
> >
> > + the designed pre-trained tasks DIM and DID are also simple, DID is similar to InfoNCE
> > **ANS:** We'd like to spell out that our DIM and DID components are not trivial applications of existing techniques but are dedicatedly formulated for dynamic information modelling. Our DID contrasts between source code and its concatenation with test cases instead of different transformed copies of the same samples in conventional InfoNCE formulations. We carried out comprehensive experiments to justify our designs. To be concrete, we tried replacing our DID with code-to-code or code-to-test cases contrastive learning, both of which share a similar formulation as InfoNCE and showed inferior performance to our FuzzPretrain equipped with DID. By comparing to those variants, we'd like to demonstrate that simply applying existing code modelling techniques to test cases is ineffective in exploring dynamic program information.
> >
> > W3: more code-related tasks are need to evaluate
> > **ANS:** We referred to CodeXGLUE as the standard benchmark involving diverse code-related tasks and evaluated on most of the code understanding tasks that are commonly adopted in previous works and have publicly accessible testing data. We are more than glad to perform experiments on more code-related tasks if specific suggestions can be given.
> >
> > References
> > [1] Eui Chul Shin, Illia Polosukhin, and Dawn Song. Improving neural program synthesis with inferred execution traces. NeurIPS, 31, 2018
> > [2] Xinyun Chen, Dawn Song, and Yuandong Tian. Latent execution for neural program synthesis beyond domain-specific languages. NeurIPS, 34: 22196–22208, 2021
> > [3] Yangruibo Ding, Ben Steenhoek, Kexin Pei, Gail Kaiser, Wei Le, and Baishakhi Ray. Traced: Execution-aware pre-training for source code. In ICSE, 2023
> > [4] Kaiming He, Haoqi Fan, Yuxin Wu, Saining Xie, and Ross Girshick. Momentum contrast for unsupervised visual representation learning. In CVPR, pp. 9729–9738, 2020.
> > [5] Chenxiao Liu, Shuai Lu, Weizhu Chen, Daxin Jiang, Alexey Svyatkovskiy, Shengyu Fu, Neel Sundaresan, and Nan Duan. Code execution with pre-trained language models. ACL, 2023
> > [6] Daya Guo, Shuai Lu, Nan Duan, Yanlin Wang, Ming Zhou, and Jian Yin. Unixcoder: Unified cross-modal pre-training for code representation. In ACL, 2022

---

> ### Author Response · Authors · 2023-11-22
> **Any further questions or comments?**
>
> Dear Reviewer EByC,
>
> We are truly grateful for your review and hope that our responses have effectively addressed your concerns. As the final deadline for reviewer-author discussions draws near, we are eager to receive any additional feedback you may have. We are more than willing to offer further clarification and engage in a deeper conversation if any of your concerns remain. Many thanks!
>
> Best regards,
> Authors

---

### Official Review · Reviewer_LioK · 2023-11-06

**Soundness:** 3 good
**Presentation:** 1 poor
**Contribution:** 2 fair
**Rating:** 5
**Confidence:** 4

**Summary:**

This paper proposed FuzzPretrain, a new pretraining method for learning code representations that incorporates program execution semantics. More specifically, FuzzPretrain uses fuzzing to produce test cases for programs and then incorporate a binary classification objective to match the programs and the corresponding test cases. In addition to this object, MLM and contrastive learning methods are also used during pretraining. Experiments are conducted for code-to-code search, clone detection, defect detection and text-to-code search tasks. And results show that the proposed method is able to outperform recent works that uses similar-sized models.

**Strengths:**

S1: This paper explores a very important domain, as a reliable, semantic-aware code embedding could be helpful for many code-related tasks;
S2: This paper proposed an interesting method that combines representation learning and code execution via fuzzing, and provides a simple way of incorporating the test cases into the pretraining tasks;
S3: The ablation studies are rather complete for the readers to understand the contribution of each part in the pretraining objectives.

**Weaknesses:**

W1: The presentation of this work is quite poor. More specifically:
* The notations are a bit (unnecessarily) complex (e.g., as a matter of fact, you can hardly find a symbol without any super/sub-script). An example is with Equation 2, I don't think it is necessary to write down the formulation of a linear layer for binary classification, nor the cross-entropy loss;
* Table 1 is a bit hard to parse, I assume they are some kind of "transfer" between different programming languages (nothing is mentioned in the caption), but even with that assumption, there are many questions needed to be answered;

W2: Some of the experiment settings are questionable. See the "questions" section for details;
W3: Compared with previous work, the improvements are quite marginal (i.e., 1 point or less) on 2 of the 3 reported tasks. Also, it would be great to mark the parameter count in table 3.

**Questions:**

Q1: In table 1, why is the first row in grey? Same question with table 2;
Q2: For table 1, what does the languages in the first and second row mean? What about C/C++, in the first line of Section 4, it was mentioned that C/C++ is also part of the training data.
Q3: Still table 1, is "w/o DIM" and "w/o DID" rows accumulative? If not, is "w/o DIM + w/o DID" the same as MLM variants in the table?
Q4: Can you explain why you alternate the three objectives between different mini-batches, and not simply add them or interpolate between them?
Q5: Since the model is both trained and tested on CodeNet, can you elaborate on how you split the data and if some precautions are taken to prevent implicit data leakage?
Q6: Can you give a hypothesis on why the effect of DIM and DID objectives are vastly different for CodeBERT and UniXcoder models (see Table 1)
Q7: Do you plan to compare with some LLM-based embedding methods? E.g., embedding models from OpenAI.
Q8: What does the color entail in Figure 3? It was mentioned that only a subset of the points are picked to be colored, how do you pick this subset? My concern is that if you pick the ones that are closely clustered in Fig 3(b) and visualize the same points in Fig 3(c), it will look scattered as well. Thus I don't think this (the colored points) shows if one embedding is better than the other. However, the groupings do look better spaced than UniXcoder.

---

> ### Author Response · Authors · 2023-11-20
> **Responses to Reviewer LioK**
>
> Q1: In table 1, why is the first row in grey? Same question with table 2.
> **ANS:** The first rows in Tables 1&2 are marked in grey because they are not directly comparable to the other rows in the same tables as training on different data. They were reported to demonstrate that we are able to obtain substantial improvements over our starting points even though our continual training was conducted on a smaller and less diverse code corpus.
>
> Q2: For table 1, what does the languages in the first and second row mean? what about C/C++
> **ANS:** The first and second rows in the header indicate the programming languages of the query and the candidate code snippets in the code-to-code retrieval (search) task, respectively. We followed the setting of this task introduced in UniXcoder to evaluate on the three reported languages. We elaborated on this in the revised caption of Table 1. Besides, we further evaluated on 50 randomly held-out coding problems in the C++1400 split of CodeNet. Our FuzzCodeBERT yielded 68.65% mAP while CodeBERT and CodeBERT-MLM achieved 13.22% and 24.08%, respectively.
>
>
> Q3: Still table 1, is "w/o DIM" and "w/o DID" rows accumulative? If not, is "w/o DIM + w/o DID" the same as MLM variants in the table?
> **ANS:** No, they are not accumulative. And yes, "w/o DIM + w/o DID" is the same as the MLM variant. Table 1 has been refined to clarify this.
>
> Q4: Can you explain why you alternate the three objectives between different mini-batches, and not simply add them or interpolate between them?
> **ANS:** We alternated between different objectives for the concerns of efficiency and GPU consumption. The three training objectives require inputs in different forms, hence, three forward passes are needed for their joint optimisation. This will lead to accumulated GPU consumption and refrain us from fitting the desired batch size into GPUs. Such an alternative training strategy has also been used in previous works like UniXcoder and GraphCodeBERT.
>
> Q5: Since the model is both trained and tested on CodeNet, can you elaborate on how you split the data and if some precautions are taken to prevent implicit data leakage?
> **ANS:** We adopted the testing set provided by UniXcoder and the benchmark datasets provided in CodeNet for training. Given that the testing set involves more than 70% of the coding problems in CodeNet, it is challenging to exclude them from training data. Therefore, we consider the searching of those overlapping samples as transductive inference problems. This is also practical given that the training data of the latest code models cover a large proportion of the open-source projects in Github and is likely to involve the code-of-interests to users.
> As suggested, we have further evaluated in an inductive setup where the query and the candidate code snippets are submissions to 50 random coding problems of each programming language that have never been seen during pre-training. The header in the below table indicates the data splits provided by CodeNet. As can be seen, the superiority of our FuzzPretrain holds.
> That is, these results show that our model is effective not only in the transductive inference setup for code search, but also in an inductive setup where no training/test overlap exists. These discussions have been added to the Appendix A.2 of the paper.
>
> |              | C++1000 | C++1400 | Py800 | Java250 | Overall |
> |--------------|---------|---------|-------|---------|---------|
> | CodeBERT     | 13.95   | 13.22   | 31.23 | 26.72   | 21.28   |
> | CodeBERT-MLM | 26.34   | 24.08   | 48.71 | 34.94   | 33.52   |
> | FuzzCodeBERT | 69.98   | 68.65   | 78.13 | 69.98   | 71.89   |

---

> > ### Author Response · Authors · 2023-11-20
> > **Responses to Reviewer LioK**
> >
> > Q6: Can you give a hypothesis on why the effect of DIM and DID objectives are vastly different from CodeBERT and UniXcoder models (see Table 1)
> > **ANS:** Our hypothesis is that there is a tradeoff between forgetting prior knowledge and acquiring new knowledge during continual learning of a base model. The new knowledge obtained by training with DIM or DID on the new data can easily outweigh the prior knowledge held by a simpler base model like CodeBERT and lead to improvements. On the contrary, a stronger base model is more likely to degrade after continual learning with objectives that are different from and simpler than the ones it adopted and on new data. This is also supported by applying a conventional MLM objective to both the base models in Table 1. This could possibly be relieved if continual training with the same objectives in combination with our new designs. However, since the training implementations of UniXcoder are not publicly available, we did not get the chance to fully testify this.
> > Luckily, even though DIM and DID contribute differently to CodeBERT and UniXcoder, when they are adopted jointly, we still observe consistent performance gains with both CodeBERT and UniXcoder models. Discussions have been added to the first paragraph of Sec 4.1.
> >
> > Q7: Do you plan to compare with some LLM-based embedding methods? E.g., embedding models from OpenAI.
> > **ANS:** This is a good suggestion, we will further explore the effectiveness of our FuzzPretrain model by combining it with the larger models to be compared with the LLM suggested.
> >
> > Q8: What does the color entail in Figure 3? It was mentioned that only a subset of the points are picked to be colored, how do you pick this subset?
> > **ANS:** Samples of the same colours in Figure 3 are code submissions to the same coding problems in CodeNet, which are considered functionally equivalent. These samples are illustrated to compare the representational space of UniXcoder and our FuzzUniXcoder. We randomly selected 10 out of 50 clusters to be coloured and kept them consistent for both the models in sub-figure (b) and (c). Only 10 clusters were highlighted because marking all the clusters with different colours will make them less visually distinguishable.
> >
> > W1: The presentation of this work is quite poor
> >
> > + The notations are a bit (unnecessarily) complex
> > **ANS:** The superscripts were used to highlight different types of inputs and the subscripts were only used to indicate the index of elements in a sequence. We take either source code or test cases or their concatenations as inputs and each of them has its own tokens and features. In this case, superscripts were useful to tell them apart. We have followed the suggestion to simplify our notations by denoting source code, test cases and their concatenation as S, D and H so that the superscripts can be more intuitive. Besides, we also summarised all the notations along with their definition in Table 4 in Appendix A.1 to make the notations clearer.
> >
> > + Table 1 is a bit hard to parse
> > **ANS:** Please see Q1-Q3 answered above, the caption of Table 1 has been revised for elaboration.
> >
> > W2: Some of the experiment settings are questionable
> > **ANS:** Please refer to Q1, 2, 3, 5, 6, 8 answered above.
> >
> > W3: the improvements are quite marginal. it would be great to mark the parameter count in table 3
> > **ANS:** The effectiveness of our models is well supported by the non-negligible improvements we made to the models that were trained on the same data as ours as shown in Tables 1&2 (e.g. our FuzzCodeBERT yielded 6%+ improvements on code search, 4%+ on clone detection, and 1.7% on text search). Meanwhile, the results in Table 3 imply that our model remained competitive with existing works even though we are trained on a much smaller and less diverse code corpus (discussed in the updated Appendix A.3) than their CodeSearchNet.
> > It is possible to obtain further gains with our FuzzPretrain by conducting fuzzing for complete projects so as to collect test cases for isolated functions on CodeSearchNet.
> > As for the parameter count, in fact, all the models being compared share a similar number of parameters (125M) by adopting the same BERT-style architecture as the backbone network. We have revised the caption of Table 3 to emphasise this fact to avoid possible confusion.

---

> > > ### Author Response · Authors · 2023-11-21
> > > **Responses to Reviewer LioK**
> > >
> > > In terms of Q7, we further conducted a preliminary comparison to the LLMs from OpenAI by using their "text-embedding-ada-002" model for python-to-python code search in CodeNet. The OpenAI's model yielded 35.91% mAP while ours are 30.47% and 47.77% when adopting either CodeBERT or UniXcoder as the base model, respectively. Given that CodeNet carefully removed near-duplicated submissions to the same coding problems with over-high syntactic similarity, such initial evaluation results indicate that semantic code search is fundamentally challenging and the test cases we adopted are strong indicators of program's functionality, which ensure our competitiveness to the larger and more complex models.

---

> ### Author Response · Authors · 2023-11-22
> **Any further questions or comments?**
>
> Dear Reviewer LioK,
>
> We are truly grateful for your review and hope that our responses have effectively addressed your concerns. As the final deadline for reviewer-author discussions draws near, we are eager to receive any additional feedback you may have. We are more than willing to offer further clarification and engage in a deeper conversation if any of your concerns remain. Many thanks.
>
> Best regards,
> Authors